# Changes in Body Composition and Physical Performance after a Six-Week International Tour in Young Chilean Female Tennis Players

**DOI:** 10.3390/sports12030078

**Published:** 2024-03-11

**Authors:** Pablo Luna-Villouta, Luis Valenzuela Contreras, Cristian Martínez Salazar, Jorge Flández Valderrama, Carlos Matus-Castillo, Carol Flores-Rivera, Marcelo Paredes-Arias, Rodrigo Vargas-Vitoria

**Affiliations:** 1Facultad de Educación, Departamento de Educación Física, Universidad de Concepción, Concepción 4030000, Chile; pabloluna@udec.cl; 2Facultad de Educación, Pedagogía en Educación Física, Universidad Católica Silva Henríquez, Santiago 8330226, Chile; lvalenzuela@ucsh.cl; 3Departamento de Educación Física, Deportes y Recreación, Universidad de La Frontera, Temuco 4780000, Chile; cristian.martinez.s@ufrontera.cl; 4Facultad Filosofía y Humanidades, Instituto de Ciencias de la Educación, Escuela de Educación Física, Universidad Austral de Chile, Valdivia 5090000, Chile; jflandez@uach.cl; 5Departamento de Ciencias del Deporte y Acondicionamiento Físico, Universidad Católica de la Santísima Concepción, Concepción 4030000, Chile; cmatus@ucsc.cl; 6Facultad de Educación y Ciencias Sociales, Universidad Andres Bello, Concepción 4030000, Chile; carol.flores@unab.cl; 7Escuela de Salud, Técnico Superior en Preparación Física, Instituto Profesional Duoc UC, Puente Alto 8190777, Chile; marcelo041192@gmail.com; 8Facultad de Ciencias de la Educación, Pedagogía en Educación Física, Universidad Católica del Maule, Talca 3460000, Chile

**Keywords:** body composition, physical performance, women, young boys, tennis, tour

## Abstract

Junior tennis players travel a lot to play tennis tournaments; this causes them to spend a lot of time away from their homes and disrupts their training, which could reduce their performance and increase the risk of injury. The purpose of this study was to analyze the changes in physical performance and body composition after a six-week international tour in young Chilean female tennis players. Thirty young female tennis players (15.4 ± 0.6) participated in this study. Body weight, skinfolds, and perimeters were measured. Body fat percentage (BFP) and skeletal muscle mass (SMM) were calculated. For physical performance, 5 m sprint, the 505 with stationary start test (505 test), the pro-agility test, countermovement jump (CMJ), and medicine ball throw (MBT) were evaluated. The results show a significant increase in BFP and decrease in SMM (*p* < 0.01; d = −0.18 and 0.19, respectively). In terms of physical performance, 5 m sprint, the 505 test (*p* < 0.01; d = −0.95 and −0.95, respectively), CMJ, MBT, and HJ significantly decreased post-tour (*p* < 0.05; d = 0.96, 0.89 and 0.47, respectively). We conclude that, after a six-week international tour, there were changes in body composition and a significant decrease in 5 m sprint, the 505 test, CMJ, and MBT.

## 1. Introduction

Tennis is a complex and intermittent sport which is characterized by a high demand for physical factors, such as strength, power, agility, and speed, as well as technical skills related to serving and groundstrokes [1,2]. The typical mean duration of a three-set match is 1.5 h, with an effective playing time of 30%, executing 300 to 500 high-intensity movements [3,4]. In this regard, the analysis of these factors is decisive in differentiating successful players from others and particularly relevant for training purposes [5,6].

It has been reported that the current population of junior competitors is twice the size of the professional tour [4], with a calendar of a high number of competitions in tournaments of different national and international levels [7]. The above specifies the relevance of specific physical conditioning preparation in accordance with the demands of tennis in the junior category [8]. Additionally, elite tennis players travel extensively for tennis tournaments to gain or maintain ranking points, which invariably interrupts the training process [9]. It is common that throughout the competition season, young tennis players are away from local facilities for several weeks and without direct supervision of their trainers, which could reduce the effects of training and their performance [10] and increase the risk of injury [1,11,12]. This occurs because the demands of tennis tournaments vary depending on the matches won, the court surface (i.e., grass, clay, and hard courts), and the type of matches played (i.e., qualifying, main draw, doubles). In addition, travel and days off between tournaments should be considered [9], and at the junior level (i.e., U16, U14), players usually take part in two consecutive tennis matches per day (i.e., morning and afternoon sessions) as part of their competition schedule [7]. Therefore, it is highlighted that it is very important to collate information about changes in body composition and physical fitness on tours that involve departures of several weeks [1,9].

In tennis, studies on the changes or effects of prolonged tours on body composition and physical and technical performance are still scarce. Murphy et al. [9] reported that a four-week tennis tour decreases performance in 5, 10, and 20 m sprint (d = 0.50–0.70); however, there were also small decreases in agility, CMJ, and aerobic capacity (d = −0.41–0.23). Kovacs et al. [10] evaluated the impact of a five-week structured, but unsupervised, break in collegiate tennis players, concluding that a 5-week interruption of normal training can result in significant reductions in speed, power, and aerobic capacity. Luna-Villouta et al. [1] analyzed the effects of a six-week international tour on physical performance and body composition in young tennis players, determining that BFP increases and SMM decreases significantly (*p* < 0.05); in addition, agility and 5 and 10 m sprints decrease significantly (*p* < 0.05; d = −0.63 and 1.10). However, there were no significant changes in CMJ and MBT (*p* > 0.05). These studies [1,9,10], with the exception of Murphy et al. [9], tended to focus on male tennis players, so women remain under-represented in the evidence base [13]; as a result, information collected from male athletes is often applied to the female population [14]. However, given the biological differences between men and women, applying studies conducted in male athletes to female athletes appears to be inappropriate [15].

Current training and nutrition planning for women is planned with models adapted to males without adaptation to the physiological needs demanded by the female sex [16]. Sex-related differences in muscle mass and fat mass are evident, but how sex hormones act to regulate energy metabolism and protein turnover during adolescence remains an enigma [17]. Previous studies have described that puberty favors a rapid increase in fat mass, relative to other tissues, such as muscle mass [18,19]. Hormones may influence tissue composition, function, and the risk of injury [17]. The hormonal profile has implications for adaptation to physical and athletic training [20,21,22]; estrogen has an anabolic effect on muscles by reducing protein turnover and improving sensitivity to resistance training [17]. The influence of the menstrual cycle on performance is one of the most controversial issues in research in women’s sport because the results are contradictory and fail to demonstrate with complete precision that hormonal variation during the menstrual cycle influences markers of physical performance [22]. Therefore, it is necessary to individually monitor the menstrual cycle and its associated symptoms [21].

Thus, the hypothesis of this research is that a six-week international tour significantly affects physical performance and body composition in young female tennis players.

Therefore, the main objective of this study was to analyze the changes in physical performance and body composition after a six-week international tour in young Chilean female tennis players. Furthermore, as a secondary objective, we sought to determine the influence of national ranking on the changes in physical performance and body composition after the six-week international tour, for which two groups were established: “national ranking 1–20” (players ranked between 1 and 20) and “national ranking 21–45” (players ranked between 21 and 45).

## 2. Materials and Methods

### 2.1. Participants

This study was observational, cross-sectional, and used descriptive characteristics. The sample was non-probabilistic for convenience. Thirty young female tennis players of Chilean nationality, between 14 and 16 years old, participated voluntarily. Originally, the top fifty players in the Chilean Junior Women’s Tennis Ranking were invited; twenty players did not participate for different reasons (non-response, injuries, travel, etc.). The evaluations were carried out a week before (pre-tour) and a week after (post-tour) a six-week international tour of the Confederation of South American Tennis (COSAT); the tournaments were played in Ecuador (1 week), Argentina (1 week), Brazil (2 weeks), Colombia (1 week), and Peru (1 week). The tennis players played 4 to 7 matches per tournament (mean = 5.5); all tournaments were played on clay tennis courts.

All participants belonged to tennis clubs in Santiago, Chile. The inclusion criteria were as follows: (1) declare female gender; (2) be between 14 and 16 years old; (3) compete in international tournaments in the last 18 months; (4) classification between number 1 and 50 in the ranking of the Chilean Tennis Federation (FETECH) according to their age; (5) weekly training volume of at least 15 h; and (6) complete the six-week COSAT international tour. The exclusion criteria were as follows: (1) not completing the evaluations and (2) presenting an injury that affected the results of the evaluations, which was reported by the player or coaching staff. G*Power version 3.1.9.7 (Heinrich Heine Universität Düsseldorf, Düsseldorf, Germany) was used to process the sample running a single-sample *t*-test which showed a bilateral alpha error of 5%, effect size 0.50, and statistical power of 80%.

For the data collection process, first, authorization was requested from the directors of the tennis clubs through a letter describing the objective and the tests to be carried out. Second, consent forms were given to the parents of each player, informing them about the objective and the characteristics of the research. Third, the participation of the players was confirmed by signing an assent, in accordance with the Declaration of Helsinki for research in humans [23]. Furthermore, a competent ethics committee in the academic field (Ethics Committee of the Universidad San Sebastián, Chile; USS 51-2018-20; date 9 January 2019) approved the study.

### 2.2. Procedures

Data collection was carried out in the morning, before training, with an ambient temperature between 11° and 15° Celsius. All measurements were carried out in the sports clubs during normal training hours. An experienced evaluator, applying the protocols described by Marfell-Jones et al. in 2012, carried out anthropometric and body composition evaluations [24]. Height (cm) was checked using a stadiometer graduated in millimeters (Seca 220, Hamburg, Germany). Body weight was verified with a mechanical scale with a range of 0 to 220 kg and a precision of 50 g (Seca 700, Hamburg, Germany). Skinfolds were measured on the right side of the tennis player with an anthropometric caliper (Harpenden^®^, Baty International Ltd., West Sussex, UK); the triceps brachial, anterior thigh, and medial leg were measured. The perimeters of the arm, thigh, and leg were also measured on the right side of the body using a metallic tape (Lufkin ^®^ Metallic, Medina, OH, USA). Measurements of each circumference were taken from the midline of the body segment being measured. Anthropometric measurements were taken twice or three times when the difference between the first and the second values was higher than 0.5 cm in height, 0.05 kg in body weight, 1% in circumferences, and 5% in skinfolds. The mean of all the measurements was used for the data analysis. The technical error of the measurements ranged from 0.30% to 0.75%.

Body fat percentage was calculated with the equation by Slaughter et al. (1988) (BFP = 0.610 (triceps + calf) + 5.1) [25]. Skeletal muscle mass (SMM) was calculated using the equation proposed by Poortmans et al. (2005) [26], SMM (kg) = height × ((0.0064 × corrected circumference for upper arm^2^) + (0.0032 × corrected circumference for thigh^2^) + (0.0015 × corrected circumference for calf^2^) + (2.56 × sex) + (0.136 × age)). Biological maturation was obtained by age at peak height velocity (APHV) using the equation by Moore et al. (2015) [27] (maturity offset = −7.709133 + (0.0042232 × (age × height)). The mean of all the measurements taken was used in all the equations.

All the physical performance tests were made on clay tennis courts after anthropometric evaluations. Participants wore competition clothing (shorts, shirt or dress, and tennis shoes). Evaluations were carried out by two experienced evaluators (Masters in Sports Sciences, MSc.) previously trained in administering the tests. Testing was conducted before and after the international tour, twice per individual, and the best results were used for the data analysis, with a 1 to 5 min break between tests. The test started with a 15 min warm-up (general physical exercises and stretching). The application was organized as follows. First, for the 5 m sprint test, girls started in a standing position, with the dominant foot behind the starting line; at the signal, they ran the indicated distance as fast as possible [28]. Second was the 505 with stationary start test (505 test); in this test, the players began by standing behind the starting line, and they accelerated at maximum speed in a straight line until they reached a line located 5 m away, where they had to pivot with one foot and turn 180° to return to the starting line departure, completing a total distance of 10 m [7]. Third, the pro-agility test was administered; the players started in a neutral position, and then they turned and ran to the side (5 m), touching a cone with their hand. Then, they turned to the opposite side and ran 10 m to the farthest cone. Finally, the player returned to the middle and the test ended when they reached the starting position [29]. In all these tests, a photoelectric cell system (Witty, Microgate^®^, Bolzano, Italy) was used to measure the best time. Fourth, the medicine ball throw was performed standing and with one hand (MBT); the throw was executed by the side of the head with a 2 kg ball and with the player’s dominant hand [30]. The distance was measured with a Stanley Power Lock millimeter tape (USA). Fifth, countermovement jumps (CMJs) were performed, and the height reached per jump was measured with a Globus Ergo Jump platform (Bosco System) according to procedures previously described [31].

### 2.3. Statistical Analysis

Statistical analysis was carried out using 17.0 SPSS IBM Corp. (IBM^®^, Somers, NY, USA). The Shapiro–Wilk test determined the normal distribution of the variables. Data are presented as mean and standard deviation (SD). The mean was calculated for the percent difference in the pre- and post-tour measurements (% Difference). The differences between pre- and post-tour measurements were determined using the t-test for related samples. In addition, the effect size (ES) of the changes in each variable was calculated with Cohen’s d, interpreted as follows: 0.2 (small), 0.5 (moderate), and 0.8 (large) [32], according to currently applied procedures for paired samples [32,33]. To complement the analysis, the t-test for related samples, the differences between the pre- and post-tour measurements were compared for the “national ranking 1–20” and “national ranking 21–45” groups. The significance level used was *p* < 0.05.

## 3. Results

Table 1 presents the mean, standard deviation (SD), *t*-test, and effect size given (Cohen’s d) for the comparison between the pre- and post-tour tests of body composition and anthropometric measurements. The post-tour values are lower in SMM and ∑3 perimeters, where these decreases are significant (*p* < 0.01), and the ES is “moderate” (d = 0.52 and 0.79); meanwhile, BW, ∑3 skinfolds (mm), and BFP (%) increase significantly post-tour (*p* < 0.01), with the ES being “moderate” (d = −0.54 to −0.74).

Table 2 contains the mean, standard deviation (SD), *t*-test, and effect size given (Cohen’s d) for the comparison between the pre- and post-tour tests of physical performance. Significant decreases are observed in 5 m sprint, the 505 test (*p* < 0.01), CMJ, and MBT (*p* < 0.05), with the ES being “large” (d = −0.95 to 0.96). In addition, in terms of HJ, there was also a significant drop in the post-tour test (*p* < 0.05) although the ES was “moderate” (d = 0.47). On the other hand, in the pro-agility test, there were no significant differences (*p* > 0.05).

Figure 1 shows the comparison of body composition and anthropometrics measures pre- and post-tour for the “national ranking 1–20” and “national ranking 21–45” groups. The groups present a significant decrease post-tour (*p* < 0.05) in BW, ∑3 Skinfolds, BFP, ∑3 perimeters, and SMM.

Figure 2 shows the comparison in the pre- and post-tour physical performance tests for the “national ranking 1–20” and “national ranking 21–45” groups. The two groups show a significant decrease in physical performance in 5 m sprint, the 505 test, CMJ, and MBT (*p* < 0.05). Furthermore, the “national ranking 21–45” group showed a significant decrease in the pro-agility test (*p* < 0.05); on the contrary, there were no significant differences in the “national ranking 1–20” group (*p* > 0.05).

## 4. Discussion

This study aimed to analyze the changes in physical performance and body composition after a six-week international tour in young Chilean female tennis players. The results show that in terms of body composition, at the end of the international tour, there was a significant increase in BW, BFP (%), and ∑3 skinfolds (*p* < 0.01), with the ES being “moderate” (d = −0.54 to −0.74); as well, a decrease in SMM and ∑3 perimeters was observed (*p* < 0.01), and the ES was “moderate” (d = 0.52 and 0.79). The variables of physical performance showed significant decreases in 5 m sprint (*p* < 0.01), the 505 test (*p* < 0.01), CMJ (*p* < 0.05), and MBT (*p* < 0.05), with the ES being “large” (d = −0.95 to 0.96), as well as HJ (*p* < 0.05; d = 0.47). Furthermore, only the “national ranking 21–45” group showed a significant decrease in the pro-agility test (*p* < 0.05).

The post-tour tests demonstrated a significant decrease (*p* < 0.01; d = −0.95) in the performance of the 5 m sprint test; this decrease coincides with what was reported in a 5-week tour in Australian tennis players [9], although the values of the Australian tennis players were lower than those generated in our research (d = −0.70). These results also coincide with the decrease detected after a six-week international tour in young Chilean male tennis players [1], although the effect size was greater in men (d = −1.10). Such detriments to speed may be a consequence of a lack of specific training, that is, a limited short duration and maximal sprint training, resulting in the poor adaptation of acceleration characteristics [9]. These results suggest that continued match play may not provide enough training stimulus for speed, and when match play dominates over training exercises for extended periods, speed capacities can suffer decreases [9,34].

The muscle power tests, CMJ, HJ, and MBT, also showed significant reductions in the post-tour measurement. Other studies with similar characteristics did not report decreases in muscle power [1,9,10]. These differences may be due to the lack of specific power training in tennis players prior to the international tour, since it has been reported that performance in these tests can be maintained for a period of up to 4–5 weeks without specific training [9,10,35,36]. At the same time, it has been reported that higher-ranked young players have higher levels of muscular power [37,38]. Furthermore, in tennis, stroke and serve techniques are commonly associated with power, so a lack of speed could potentially put you at a disadvantage against your opponent [39,40,41]. The results show that impaired strength levels need to be taken into consideration while planning match preparation and recovery before, during, and after the tournament [34].

The 505 test showed a significant decrease after the tour with a “large” ES (d = −0.95); however, in the pro-agility test, there were only significant reductions in the “national ranking 21–45” group. These results are above those previously reported by Luna-Villouta et al. [1] on a six-week tour (d = −0.63); although, in this study, we used the MAT test (Modified Agility Test), and reductions occurred in all tennis players regardless of their ranking. These limited changes in agility were also highlighted by Murphy et al. [9], who pointed out that these reductions may be due to the fact that typical movements in tennis matches (lateral and frontal displacements, jumps, turns, etc.) can provoke sufficient stimuli to agility maintenance.

When analyzing the changes in body composition, in general, lower post-tour values are observed for SMM and ∑3 perimeters, and there is an increase in BFP and ∑3 skinfolds, with the ES being “moderate” (d = −0.54 and 0.79). These values partially coincide with those of Luna-Villouta et al. [1], who detected the loss of SMM, but unlike this study, they reported a decrease in BFP and ∑3 skinfolds. In this aspect, there are several studies reporting different results [42,43,44,45], which shows the complexity of these aspects in young athletes due to the influence of maturation, growth, nutrition, and training level [46,47]. Thus, it has been detected that high levels of fat mass affect physical performance in young athletes [46,48,49], for example, in the 5 m sprint of young tennis players [50]. Therefore, it is important to consider the aforementioned aspects before diagnosing and starting any nutritional treatment or making changes in training.

The decreases in physical performance and body composition after a six-week international tour demonstrate the importance of adequate training and preparation of strength, power, speed, and agility in young female tennis players; this can help avoid losses in performance and successfully overcome the demands of competition and multi-week tours [7,9,51]. The results of the study show that during extensive tours, players must travel with an alternative training plan, with strength, power, and speed exercises, to be used on days without tournaments or matches [1,9,10]. Likewise, changes in body composition highlight the importance of players having specific nutritional guidelines for these trips, which must incorporate nutrition, hydration, and the use of recovery methods [1,7,34,52]. It is recommended that after a physically tough tournament, players need over 24 h of rest for full physical recovery [34]. Another practical recommendation is that players periodically measure their weight and perimeters with small and easily transportable materials [1]. Currently, all these recommendations and measurements are easy to monitor remotely through software applications on mobile devices, such as smartphones, laptops, and smartwatches [1].

The menstrual cycle and hormonal profile are aspects that may influence evaluations of the physical performance and body composition of young female tennis players. This influence is still being studied, due to the contradictory results of research in this area [16,17,20,21,22], but we believe that they should be monitored and considered for future research.

Despite the important and significant findings, this study is subject to some limitations. First, participants had very similar physical fitness levels and characteristics, which reduces the utilization of the results in other populations. Second, the study design did not measure the influence of other physical, biological, and psychological factors that may intervene in the results obtained. Third, the absence of a “control group” did not allow us to rule out the influence of the limiting factors previously mentioned. Among the strengths of the study, it stands out as there is scarce research of this type, especially in female tennis players. In addition, reliable procedures and physical performance tests that are reliable and quick to apply were used, making their reproduction relatively easy.

## 5. Conclusions

Based on this study’s results, we conclude that, after a six-week international tour, there were changes in body composition (increase in body fat and decrease in muscle mass) and a significant decrease in 5 m sprint, the 505 test, CMJ, and MBT; in addition, lower-ranked tennis players decreased in agility. These results show the importance of having training plans, evaluation methods, nutrition guidelines, and recovery procedures during a six-week international tour to avoid alterations in physical performance and body composition. In addition, individual monitoring is recommended due to the individual alterations each player displayed.

## Figures and Tables

**Figure 1 sports-12-00078-f001:**
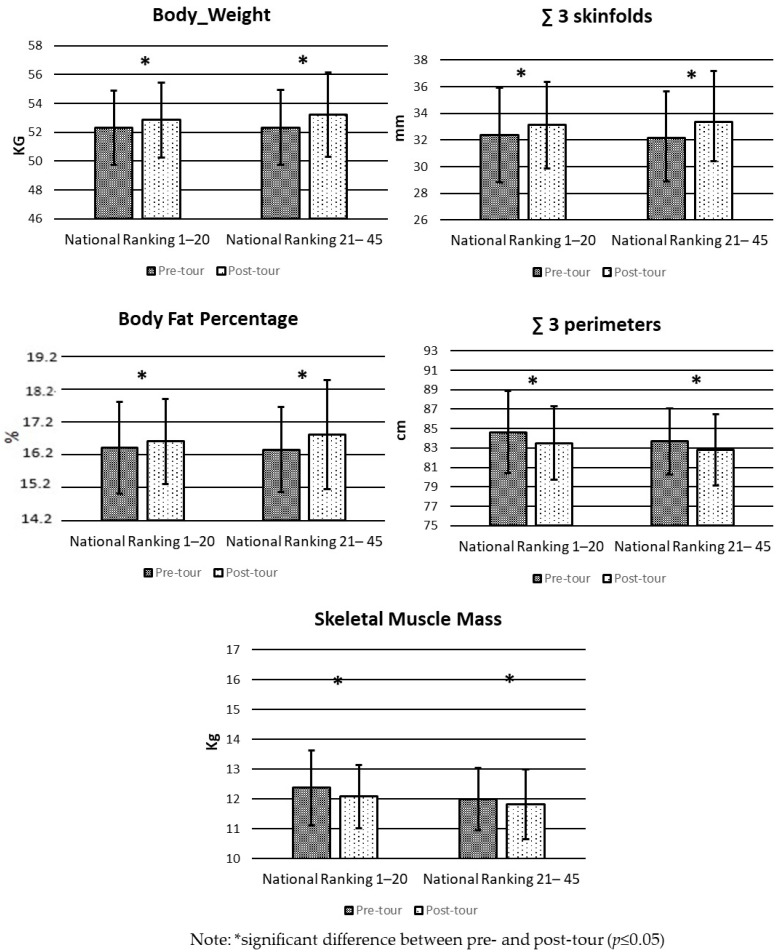
Mean and SD in pre- and post-tour body composition and anthropometric measures by Chilean women’s junior tennis ranking.

**Figure 2 sports-12-00078-f002:**
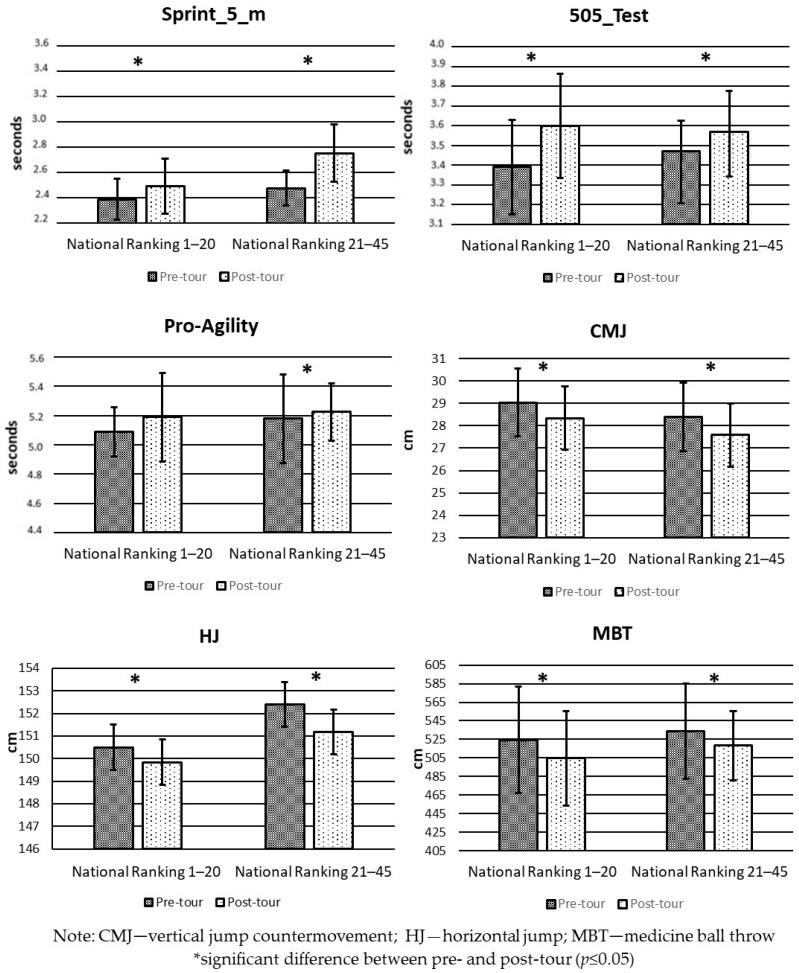
Mean and SD in pre- and post-tour physical performance tests by Chilean women’s junior tennis ranking.

**Table 1 sports-12-00078-t001:** Pre- and post-tour body composition and anthropometric measures.

Variables	Pre-Tour	Post-Tour				
	Mean ± SD	Mean ± SD	*p*-Value	% Difference	ES	
					*d*	Qualitative
Age (years)	15.4 ± 0.6	15.6 ± 0.6				
APHV (levels)	2.5 ± 0.5	2.6 ± 0.5	0.20	6.8	−0.48	moderate
Height (cm)	158 ± 3.8	158.3 ± 3.7	0.61	0.3	−0.46	moderate
BW (kg)	52.3 ± 2.7	53.0 ± 2.7	0.01 *	1.3	−0.62	moderate
∑3 perimeters (cm)	84.3 ± 3.9	83.2 ± 3.7	0.01 *	−1.2	0.79	moderate
SMM (kg)	12.2 ± 1.2	12.0 ± 1.1	0.01 *	−2.0	0.52	moderate
∑3 skinfolds (mm)	32.3 ± 3.5	33.2 ± 3.4	0.01 *	2.8	−0.74	moderate
BFP (%)	16.4 ± 1.4	16.7 ± 1.4	0.03 *	1.7	−0.54	moderate
Training hours/week	24.9 ± 3.7	24.1 ± 3.8				

Note: APHV—peak growth rate acceleration; BW—body weight; SMM—skeletal muscle mass; BFP—body fat percentage. * indicates significant difference (*p* < 0.05).

**Table 2 sports-12-00078-t002:** Pre- and post-tour physical performance tests.

Variables	Pre-Tour	Post-Tour				
	Mean ± SD	Mean ± SD	*p*-Value	% Difference	ES	
					*d*	Qualitative
Sprint_5 m (s)	2.4 ± 0.2	2.6 ± 0.3	0.01 *	6.9	−0.95	large
505 (s)	3.4 ± 0.2	3.6 ± 0.2	0.01 *	5.1	−0.95	large
Pro-agility (s)	5.1 ± 0.2	5.2 ± 0.3	0.16	1.6	−0.69	moderate
CMJ (cm)	28.8 ± 1.7	28.1 ± 1.7	0.03 *	−2.6	0.96	large
HJ (cm)	151.2 ± 7.6	150.3 ± 7.6	0.03 *	−0.7	0.47	moderate
MBT (cm)	527.9 ± 54.4	509.7 ± 46.2	0.03 *	−3.3	0.89	large

Note: CMJ—vertical jump countermovement; HJ—horizontal Jump; MBT—medicine ball throw. * indicates significant difference (*p* < 0.05).

## Data Availability

The data presented in this study are available on request from the corresponding author. The data are not publicly available due to ethical considerations of the human study.

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
