# Peer review of "Changes in Body Composition and Physical Performance after a Six-Week International Tour in Young Chilean Female Tennis Players"

_sports, 2024, doi:10.3390/sports12030078_

Round 1

Reviewer 1 Report

Comments and Suggestions for Authors

Dear Authors, I would like to express my gratitude for the opportunity to review this manuscript. It is an interesting study, but I need to make some important considerations:

-       Why didn’t you consider checking the type of dietary during the periods considered? The authors don’t think these elements could have affected the results especially of body composition?

-       The authors have not indicated the period of the menstrual cycle during performing the evaluations. An increase of body weight (BW) is typical during women's menstrual cycle, mostly due to extracellular fluid retention at menstruation days. There is evidence reporting an increase of BW in the luteal versus follicular phase, a decrease in BW in the follicular phase versus between menses (and an increase of total body water in late luteal versus follicular or early luteal phase. The authors don’ think that these elements could have affected the results obtained not only in term of body composition but also in the physical performance parameters considered?

Please add in the introduction a paragraph related to above and try to explain your results more thoroughly in the discussion also in light of the literature on the same topic.

Following, are reported some indications for the manuscript:

Line 2-4: Please, format the title considering the journal template and instructions for authors (upper and lowercase).

Point 2:

Line 5-6: Please, consider changing letters of affiliations for numbers.

Line 28:  Please, report mean and SD for age.

Line 114-115: Please, indicate the complete date of ethic approval.

Line 126-129: Please, insert more details about perimeters measurement (e.g. reference points, body position) and reference.

Line 128-129: which is the value considered for the analysis? The mean of three measurements? Please, report it.

Line 140-141: As above.

Line 157-159: Please, indicate the parameters of CMJ test considered for analysis (e.g height of jump)

Table 1: Please report in one column the mean and SD

Table 1: It’s not clear what report the column “% differences”, please could explain it?

Table 2: As above.

Figure 1: the caption of figure misses.

Figure 1: replace commas with dots in the y-axis.

Figure 1: remove the lines in the center of the figure.

Figure 1: remove the bold in “note” below the figure.

Figure 2: As above

Line 293: Please, indicate the complete date of ethic approval.

Line 301-410: Please revise the format of the references (e.g. name of the journal in italicsDOI´s not presented).

References should be described  as follows, depending on the type of work, for example

Journal Articles:

1. Author 1, A.B.; Author 2, C.D. Title of the article. Abbreviated Journal Name Year, Volume, page range.

Please check all details according author’s guidelines.

Comments on the Quality of English Language

Minor editing of English language required

Author Response

Reviewer 1:

Reply letter

Dear Reviewer 1,

Thank you againg for taking the time to review this manuscript

Point 1: Why didn’t you consider checking the type of dietary during the periods considered? The authors don’t think these elements could have affected the results especially of body composition?

Response 1: Dear reviewer, we know that nutrition is important in sports, but since we could not control it rigorously, we did not include it in the research. We hope you can understand our limitations.

Point 2: The authors have not indicated the period of the menstrual cycle during performing the evaluations. An increase of body weight (BW) is typical during women's menstrual cycle, mostly due to extracellular fluid retention at menstruation days. There is evidence reporting an increase of BW in the luteal versus follicular phase, a decrease in BW in the follicular phase versus between menses (and an increase of total body water in late luteal versus follicular or early luteal phase. The authors don’ think that these elements could have affected the results obtained not only in term of body composition but also in the physical performance parameters considered?

Response 2: Dear Reviewer, we agree with what you ask of us. In our literature review, we found research with contradictory results, which we have included in the Introduction of our paper. Please revise lines: 79-93.

Point 3: Please add in the introduction a paragraph related to above and try to explain your results more thoroughly in the discussion also in light of the literature on the same topic.

Response 3: Dear Reviewer, We have included the analysis of what is indicated. Please revise lines: 79-93, 301-304.

Point 4: Line 2-4: Please, format the title considering the journal template and instructions for authors (upper and lowercase).

Response 4: Dear Reviewer, we have reviewed and corrected what you have indicated. Please revise lines: 2-4

Point 5: Line 5-6: Please, consider changing letters of affiliations for numbers.

Response 5: Dear Reviewer, We have already corrected the affiliations numbers. Please revise lines: 5-22

Point 6: Line 28:  Please, report mean and SD for age.

Response 6: Dear Reviewer, we have incorporated your request. Please revise line 28.

Point 7: Line 114-115: Please, indicate the complete date of ethic approval.

Response 7: Dear Reviewer, we have incorporated your request. Please revise lines: 131-132

Point 8: Line 126-129: Please, insert more details about perimeters measurement (e.g. reference points, body position) and reference. Line 128-129: which is the value considered for the analysis? The mean of three measurements?. Please, report it. Line 140-141: As above.

Response 8: Dear Reviewer, we have incorporated your request. Please revise lines: 145-149. 158-159.

Point 9: Line 157-159: Please, indicate the parameters of CMJ test considered for analysis (e.g height of jump)

Response 9: Dear Reviewer, we have incorporated your request. Please revise lines: 180-181.

Point 10: Table 1: Please report in one column the mean and SD. Table 2: As above.

Response 10: Dear Reviewer, we have incorporated your request. Please revise tables 1 and 2.

Point 11: It’s not clear what report the column “% differences”, please could explain it?

Response 11: Dear Reviewer, the "%difference" column reports the mean and SD for the percentage of differences in the pre- and post-tour measurements. In the article, we have incorporated a brief explanation for this point. Please revise lines: 186-187.

Point 12: Figure 1: the caption of figure misses. Figure 1: replace commas with dots in the y-axis.

Figure 1: remove the lines in the center of the figure. Figure 1: remove the bold in “note” below the figure. Figure 2: As above

Response 12: Dear Reviewer, we have reviewed and corrected what you have indicated. Please revise figures 1 and 2.

Point 13: Line 293: Please, indicate the complete date of ethic approval.

Response 13: Dear Reviewer, we have incorporated your request. Please revise line 334.

Point 14: lease revise the format of the references (e.g. name of the journal in italics, DOI´s not presented). References should be described  as follows, depending on the type of work, for example

Journal Articles: 1. Author 1, A.B.; Author 2, C.D. Title of the article. Abbreviated Journal Name Year, Volume, page range. Please check all details according author’s guidelines.

Response 14: Dear Reviewer, we have reviewed and corrected what you have indicated. Please revise references section.

We appreciate all your comments that we believe will improve the quality and insight of our article.

Sincerely,

Reviewer 2 Report

Comments and Suggestions for Authors

·       L 65, blank missing in »5,10«

·       L 42 » The total duration of a three-set match is 1.5 hours” ? on average?

·       L 89+ Participants: missing info of response rate (how many were invited, how many refuse to participate or are missing due to other reasons)

·       L 103–7: power analysis for selected parameters gives sample size of n=13… Why do you select so large ES (0.75) for such a short 'intervention' (6 weeks)?

·       L 112: typo in 'accord-ance'

·       L 127: why do you use *metal* tape to measure perimeters and not the usual plastic or fabric?

·       L 160+: there is no information on how you test the differences between ranking groups (1-20 vs 21-45)

·       L 162: why do you check the normal distribution »of the variables”, and not the difference between pre- and post-testing (which is an actual assumption for the selected t-test)

·       L 165: there are at least three different formulas (procedures) to calculate Cohen's d. Which exactly do you use? State in the text and give a reference.

·       L 176 Table 1:

o   It seems you used wrong formula for Cohen's d (i.e. not the one for paired samples)

o   Give more precision (additional decimal place) for the 'Difference %' column, and also means/SD when needed. E.g. in height you stated the difference of 0.3%, but the actual difference between 158.3 and 158.0 is less than 0.2%.

o   Give the SD(D), i.e. SD of the *differences* between the pre- and post-testing. This statistic is more relevant than separate SD's for pre- and post- in the given context (paired samples)

·       L 186 Tables 2: same comments as for Table 1

·       Figure 1&2: you only give the significance levels (* 5%) for the pre-/post-testing *separately* (within) for each group, not for the differences *between* groups, i.e. not answering the main question here -- if there a difference of intervention effect between those two groups.

·       L 209: “(d = 0,21 a 0,27).” Always use decimal *point* (not comma)

·       L 210 and elsewhere: carefully check for typos (“decrease n 5-m”, “in fact ,”—space in front of comma etc.)

·       L 211+: “only 211 the “national ranking 21-45” group showed a significant decrease in the Pro.agility test 212 (p< 0.05).” as mentioned above, the question here is if the difference between groups is significant (not if it is significant within each of two groups separately)

·       L 267+, limitations:

o   You should add “lack of control group”: you observed the pre-post changes, but actually you don’t know to which influences these changes are attributable, besides “lack of specific physical training”, e.g. out could not rule out e.g. developmental changes, seasonal cyclic changes, post-tour tiredness or lack of motivation etc.

o   Similarly, you do not have control over nutrition and physical training (type, duration etc.) during the tour. So even if you speculate those factors may cause the observed changes, you don’t have a proof that they actually do and to what extent.

·       L 275+, Conclusions: “These results show the importance of having training plans, evaluation methods, nutrition guidelines and recovery procedures during international tours, to avoid alterations in performance and avoid injuries” -- These findings are too general and speculative. What if those athletes already have these plans, methods, guidelines etc. –which you didn’t measure or control—but *despite* of them there was ‘some’ decline in indicators (you choose). Besides, it is over speculative that those observed changes actually influence their performance and cause injuries: even if the changes are statistically significant, it’s questionable if they are ‘clinically relevant’, i.e. if they really influence performance and injuries. Aren’t those changes just ‘natural’ and influence all the participating athletes in a similar way? If they do, you can’t say they actually ‘alternate the performance’ because all athletes are affected and the relative difference between them stays similar during the tour.

Author Response

Reviewer 2:

Reply letter

Dear Reviewer 2,

Thank you very much for taking the time to review this manuscript

Point 1: L 65, blank missing in »5,10«

Response 1: Dear Reviewer, we have reviewed and corrected what you have indicated. Please revise line 65.

Point 2: L 42 » The total duration of a three-set match is 1.5 hours” ? on average?

Response 2: Dear Reviewer, the data corresponds to the typical mean duration of a match. We have incorporated this information. Please revise line 43.

Point 3: L 89+ Participants: missing info of response rate (how many were invited, how many refuse to participate or are missing due to other reasons)

Response 3: Dear Reviewer, we have incorporated your request. Please revise lines:107-109.

Point 4: L 103–7: power analysis for selected parameters gives sample size of n=13… Why do you select so large ES (0.75) for such a short 'intervention' (6 weeks)?

Response 4: Dear Reviewer, the ES used was 0.50, which gave a sample of 29 tennis players. We have incorporated this information. Sorry for our mistake. Please check line 124.

Point 5: L 112: typo in 'accord-ance'

Response 5: Dear Reviewer, we have reviewed and corrected what you have indicated. Please revise line 129.

Point 6: why do you use *metal* tape to measure perimeters and not the usual plastic or fabric?

Response 6: Dear Reviewer, according to our knowledge, the tape must be metallic so that it does not lose elasticity. If a tape made of another material is used, it must be checked very frequently. Marfell-Jones et al. 2012, Norton and Ols (1996) have reported this in their books

Point 7: L 160+: there is no information on how you test the differences between ranking groups (1-20 vs 21-45)

Response 7: Dear Reviewer, we have incorporated your request. Please revise lines:191-193.

Point 8. L 162: why do you check the normal distribution »of the variables”, and not the difference between pre- and post-testing (which is an actual assumption for the selected t-test)

Response 8: Dear Reviewer, we use the normality test because many of the statistical procedures including correlation, regression, t-tests, and analysis of variance, i.e. parametric tests, are based on the assumption that the data follows a normal distribution.

Point 9.   L 165: there are at least three different formulas (procedures) to calculate Cohen's d. Which exactly do you use? State in the text and give a reference.

Response 9. Dear Reviewer, thank you for your comments, upon review we realized that we had made a mistake when using GPOWER 3.1.9.7 to calculate the effect size, we have corrected it. The formula used is the one recommended by Faul F, Erdfelder E, Lang AG, Buchner A. G*Power 3: A flexible statistical power analysis program for the social, behavioral, and biomedical sciences. Behav Res Methods. 2007 May 1;39(2):175–91;

Point 10: L 176 Table 1: o   It seems you used wrong formula for Cohen's d (i.e. not the one for paired samples) o   Give more precision (additional decimal place) for the 'Difference %' column, and also means/SD when needed. E.g. in height you stated the difference of 0.3%, but the actual difference between 158.3 and 158.0 is less than 0.2%. o   Give the SD(D), i.e. SD of the *differences* between the pre- and post-testing. This statistic is more relevant than separate SD's for pre- and post- in the given context (paired samples)

Response 10: Dear Reviewer, we have modified the wording according to what has been indicated. In addition, the "%difference" column reports the mean and SD for the percentage of differences in the pre- and post-tour measurements (the mean was calculated for each pair of cases, pre and post-tour, which is why it does not coincide with the direct calculation of the mean in the tables). In the article, we have incorporated a brief explanation for this point. Please revise lines: 186-187. Tables 1 and 2.

Point 11: L 209: “(d = 0,21 a 0,27).” Always use decimal *point* (not comma); L 210 and elsewhere: carefully check for typos (“decrease n 5-m”, “in fact ,”—space in front of comma etc.)

Response 11: Dear Reviewer, we have reviewed and corrected what you have indicated. Please revise lines 240-241.

Point 12: Figure 1&2: you only give the significance levels (* 5%) for the pre-/post-testing *separately* (within) for each group, not for the differences *between* groups, i.e. not answering the main question here -- if there a difference of intervention effect between those two groups.

L 211+: “only 211 the “national ranking 21-45” group showed a significant decrease in the Pro.agility test 212 (p< 0.05).” as mentioned above, the question here is if the difference between groups is significant (not if it is significant within each of two groups separately)

Response 12: Dear Reviewer, in the preliminary statistical analyzes we did not find a significant difference between the groups (p > 0.05), for this reason, we decided to make the comparison of each group separately. This allowed us to answer the secondary objective of the study (”determine the influence of the national ranking on the changes in physical performance and body composition after the six-week international tour).

Point 13: L 267+, limitations: o   You should add “lack of control group”: you observed the pre-post changes, but actually you don’t know to which influences these changes are attributable, besides “lack of specific physical training”, e.g. out could not rule out e.g. developmental changes, seasonal cyclic changes, post-tour tiredness or lack of motivation etc. x.

Response 13: Dear Reviewer, we have incorporated your request. Please revise lines 310-311.

Point 14: o   Similarly, you do not have control over nutrition and physical training (type, duration etc.) during the tour. So even if you speculate those factors may cause the observed changes, you don’t have a proof that they actually do and to what extent.

 L 275+, Conclusions: “These results show the importance of having training plans, evaluation methods, nutrition guidelines and recovery procedures during international tours, to avoid alterations in performance and avoid injuries” -- These findings are too general and speculative. What if those athletes already have these plans, methods, guidelines etc. –which you didn’t measure or control—but *despite* of them there was ‘some’ decline in indicators (you choose). Besides, it is over speculative that those observed changes actually influence their performance and cause injuries: even if the changes are statistically significant, it’s questionable if they are ‘clinically relevant’, i.e. if they really influence performance and injuries. Aren’t those changes just ‘natural’ and influence all the participating athletes in a similar way? If they do, you can’t say they actually ‘alternate the performance’ because all athletes are affected and the relative difference between them stays similar during the tour.

Response 14: Dear Reviewer, we think that the tour of several weeks alters the supervision over the athlete by their coaches and family, because it is common for young people to travel in groups of various players (±10 players), under the supervision of a single coach, travel several hours, play daily games and in changing environmental conditions, which can affect their training, recovery and nutrition, so we believe it is a problem that requires evidence. This has already been reported by Murphy et al 2015; 7. Gallo-Salazar et al. 2017; Ojala et al. 2013; Luna-Villouta et al. 2023; Kovacs et al. 2007. Likewise, we have revised and refined the conclusions according to your comments. Please review the Conclusions section.

We appreciate all your comments that we believe will improve the quality and insight of our article.

Sincerely,

Round 2

Reviewer 1 Report

Comments and Suggestions for Authors

 Dear Authors, thank you for considering my comments. Below,  just a few further indications:

Table 1 and Table 2: please, remove mean ± SD from % Difference columns and leave only the absolute value.

Line 343-474: Please revise the format of the references (e.g. name of the journal in italics, DOI´s not presented).

References should be described as follows, depending on the type of work, for example, Journal Articles:

Author 1, A.B.; Author 2, C.D. Title of the article. Abbreviated Journal Name Year, Volume, page range.

Please check all details according author’s guidelines, in particular respect the punctuation as indicated for example showed above.

Comments on the Quality of English Language:

Minor editing of English language required.

Comments on the Quality of English Language

Minor editing of English language required.

Author Response

Reviewer 1:

Reply letter

Dear Reviewer 1,

Thank you again for taking the time to review this manuscript

Point 1: Table 1 and Table 2: please, remove mean ± SD from % Difference columns and leave only the absolute value.

Response 11: Dear Reviewer, we have incorporated your request. Please revise tables 1 and 2.

Point 2: Line 343-474: Please revise the format of the references (e.g. name of the journal in italics, DOI´s not presented). References should be described as follows, depending on the type of work, for example, Journal Articles: Author 1, A.B.; Author 2, C.D. Title of the article. Abbreviated Journal Name Year, Volume, page range.

Please check all details according author’s guidelines, in particular respect the punctuation as indicated for example showed above.

Response 2: Dear Reviewer, we have reviewed and corrected what you have indicated. Please revise references section. All changes are marked with red letter.

We appreciate all your comments that we believe will improve the quality and insight of our article.

Sincerely,
